# Bacterial Outer Membrane Vesicles as a Platform for the Development of a Broadly Protective Human Papillomavirus Vaccine Based on the Minor Capsid Protein L2

**DOI:** 10.3390/vaccines11101582

**Published:** 2023-10-11

**Authors:** Silvia Tamburini, Yueru Zhang, Assunta Gagliardi, Gabriele Di Lascio, Elena Caproni, Mattia Benedet, Michele Tomasi, Riccardo Corbellari, Ilaria Zanella, Lorenzo Croia, Guido Grandi, Martin Müller, Alberto Grandi

**Affiliations:** 1Department of Cellular, Computation and Integrative of Biology (CIBIO), University of Trento, Via Sommarive 9, 38123 Trento, Italy; silvia.tamburini@unitn.it (S.T.); michele.tomasi.2@unitn.it (M.T.); riccardo.corbellari@unitn.it (R.C.); ilaria.zanella@unitn.it (I.Z.); lorenzo.croia@unitn.it (L.C.); 2German Cancer Research Center (DKFZ), Im Neuenheimer Feld 242, 69120 Heidelberg, Germany; yueru.zhang@dkfz-heidelberg.de (Y.Z.); martin.mueller@dkfz-heidelberg.de (M.M.); 3Toscana Life Sciences Foundation, Via Fiorentina 1, 53100 Siena, Italy; a.gagliardi@toscanalifesciences.org (A.G.); g.dilascio@toscanalifesciences.org (G.D.L.); e.caproni@toscanalifesciences.org (E.C.); m.benedet@toscanalifesciences.org (M.B.); a.grandi@toscanalifesciences.org (A.G.); 4BiOMViS Srl, Via Fiorentina 1, 53100 Siena, Italy

**Keywords:** outer membrane vesicle (OMV), proteome minimized *E. coli*, human papillomavirus (HPV), minor capsid protein L2, OMV engineering, HPV OMV based vaccine, broadly protective vaccine

## Abstract

Human papillomaviruses (HPVs) are a large family of viruses with a capsid composed of the L1 and L2 proteins, which bind to receptors of the basal epithelial cells and promote virus entry. The majority of sexually active people become exposed to HPV and the virus is the most common cause of cervical cancer. Vaccines are available based on the L1 protein, which self-assembles and forms virus-like particles (VLPs) when expressed in yeast and insect cells. Although very effective, these vaccines are HPV type-restricted and their costs limit broad vaccination campaigns. Recently, vaccine candidates based on the conserved L2 epitope from serotypes 16, 18, 31, 33, 35, 6, 51, and 59 were shown to elicit broadly neutralizing anti-HPV antibodies. In this study, we tested whether *E. coli* outer membrane vesicles (OMVs) could be successfully decorated with L2 polytopes and whether the engineered OMVs could induce neutralizing antibodies. OMVs represent an attractive vaccine platform owing to their intrinsic adjuvanticity and their low production costs. We show that strings of L2 epitopes could be efficiently expressed on the surface of the OMVs and a polypeptide composed of the L2 epitopes from serotypes 18, 33, 35, and 59 provided a broad cross-protective activity against a large panel of HPV serotypes as determined using pseudovirus neutralization assay. Considering the simplicity of the OMV production process, our work provides a highly effective and inexpensive solution to produce universal anti-HPV vaccines.

## 1. Introduction

Outer membrane vesicles (OMVs) are closed spheroid particles, 50 to 300 nm in diameter, generated through a “budding out” of the outer membrane of Gram-negative bacteria. They exert a multitude of functions, including inter and intra-species cell-to-cell crosstalk, biofilm formation, genetic transformation, defense against host immune responses, and transport of toxins and virulence factors [1].

Although discovered more than 60 years ago, OMVs have attracted attention only recently, as the scientific community realized their potential as a vaccine platform owing to their excellent adjuvanticity [2], the possibility of being engineered with heterologous antigens [3,4,5,6], and the simplicity of their production process [7]. OMV-based vaccines have already reached the market [8,9], while others are in advanced clinical phases [10,11]. 

To develop cost-effective vaccines against infectious diseases and cancer, our laboratories have been working on a new vaccine platform based on engineered OMVs from non-pathogenic *Escherichia coli* (*E. coli*) derivatives over the last few years. In particular, using Synthetic Biology, we recently created a novel *E. coli* strain, named *E. coli* BL21(DE3)Δ60, releasing OMVs_Δ60_ (herein after OMVs) deprived of 60 endogenous proteins. By eliminating endogenous proteins, we managed to improve the number of heterologous antigens that can be loaded in the OMVs, with an overall beneficial effect on the immunogenicity of the target antigens [12]. Moreover, we set up an engineering strategy that envisages the expression of foreign antigens as lipoproteins. Lipidated antigens not only can accumulate in the vesicular compartment at concentrations as high as 20–30% of total OMV proteins [6,13] but they are also often exposed on the OMV surface. This is an unexpected result considering that essentially all natural *E. coli* lipoproteins (approximately 100 as predicted by genome analysis) face the periplasmic compartment, anchored to either the inner or the outer membrane [13]. The platform has been applied to develop a number of candidate vaccines against bacterial and viral pathogens, including *Staphylococcus aureus* [6,14], group A Streptococcus (Gagliardi et al., in preparation), and SARS-CoV-2 [15].

Human papillomaviruses (HPVs) are small non-enveloped double-stranded DNA viruses with a genome size of approximately 8 kb. In the presence of microlesions in the skin, genital organs, and oropharyngeal areas, the two capsid proteins L1 and L2 attach to receptors of the basal epithelial cells and promote the entry of the virus, which starts to replicate [16]. Infection with certain HPV types can lead to disruption of the cell cycle regulation and promote prolonged host cell life, genomic instability, and cancer [17].

More than 200 different HPVs are known to infect human mucosa and skin [18]. Of these, at least 15 HPV types are associated with cancer of the anogenital and oral epithelium [19,20,21]. Most sexually active women become exposed to HPV during their lifetime [22,23], and approximately 10% of them develop a persistent infection accompanied by low- and high-grade intraepithelial lesions, the main risk factor for the development of cervical cancer [24,25]. HPV is the most common cause of cervical cancer, with more than 500,000 cases and 250,000 deaths every year.

Fortunately, several preventive HPV vaccines are available (Cervarix, Gardasil, Gradasil-9, Cecolin, Walrinvax, and Cervavax). Of those, the bivalent HPV-16/18 Cervarix^®^, the quadrivalent HPV-6/11/16/18 vaccine Gardasil^®^, and Gardasil^®^9, a nonavalent HPV-6/11/16/18/31/33/45/52/58 vaccine, are most widely used. All of them are based on the L1 protein and take advantage of the fact that when expressed in yeast or insects, the L1 protein self-assembles and forms virus-like particles (VLPs) that are morphologically and antigenically highly similar to native virions [26]. Although very effective, these vaccines are accompanied by some limitations. First, they prevent infection against the vaccine strains, but they are generally HPV type-restricted with limited cross-protection. Second, VLP vaccines are relatively unstable and require a cool chain for their distribution. Third, vaccine manufacturing is not straightforward and their production costs represent a hurdle for broad vaccination campaigns in low-income countries.

L2-based vaccines represent an alternative strategy. In particular, the N-terminal region of L2 spanning from amino acids 19 to 37 or 20 to 38 (depending on the HPV serotype sequence annotations) harbors a major cross-neutralization epitope. Interestingly, such epitope is buried in mature virions but becomes exposed and thereby accessible to antibodies during the process of virus entry [26]. In the last few years, vaccine candidates based on strings of L2 epitopes selected from different HPV serotypes fused to bacterial thioredoxin (Trx) have been proposed [27,28,29,30,31,32,33]. In particular, the Trx derived from the thermophile archaeon *Pyrococcus furiosus* (*Pf*Trx) was used to produce a fusion vaccine composed of a polytope consisting of eight L2 epitopes derived from serotypes 16, 18, 31, 33, 35, 6, 51, and 59. The vaccine candidate was shown to elicit broadly neutralizing anti-HPV antibodies [34].

In this study, we tested whether OMVs could be successfully decorated with similar L2 polytopes and whether these could induce functional antibodies. We show that OMVs are excellent carriers for L2 epitopes, which can be expressed on the surface of the OMVs as long polytopes. In particular, we show that a polypeptide composed of the L2 epitopes from serotypes 18, 33, 35, and 59 provided broad cross-protective activity as assessed using pseudovirus neutralization assay. Such OMV-based vaccine is a novel effective vaccine, whereby the simplicity of its production process represents an effective and inexpensive solution to universal anti-HPV vaccination campaigns.

Altogether our data confirm the flexibility of the OMV vaccine platform, which can be exploited for developing a broad range of vaccines, including vaccines against bacterial and viral pathogens and cancer vaccines. Moreover, considering the simplicity of the OMV production process our work provides an effective and inexpensive solution to universal anti-HPV vaccination campaigns.

## 2. Materials and Methods

### 2.1. Bacterial Strains and Cultures

*E. coli* HK100 strain was used for cloning experiments using the polymerase incomplete primer extension (PIPE) method [35]. The newly generated plasmids of interest were transformed in the *E. coli* BL21(DE3)Δ60 strain [12]. For each recombinant strain, a stock preparation (Master Seed) in Luria–Bertani medium (LB) (Sigma-Aldrich, St. Louis, MO, USA) containing 15% glycerol was prepared from an overnight (ON) culture and stored at −80 °C. Bacteria were grown in LB and when required ampicillin or chloramphenicol was added to a final concentration of 100 μg/mL and 30 μg/mL, respectively.

### 2.2. Cloning of the L2 Epitopes

Appendix A shows the strategy used to clone the L2 epitopes fused at the N-terminal domain of factor H binding protein from *N. meningitidis* (*Nm*fHbp). All the primers used for the PCRs were purchased from Metabion (Planegg, Germany) and they are summarized in Appendix A. Briefly, plasmid pET_Nm-fHbpDomA_MUC1 [13], coding the MUC1 peptides fused to the C-terminus of the domain A of fHbp (amino acids 1–120), was PCR-linearized with the primers pET_F and MUC3x_R (Appendix A). The synthetic genes coding for the L2 epitopes (Appendix A) (GeneArt, ThermoFisher, Waltham, MA, USA) were inserted into the linearized plasmid using the PIPE method. In particular, each synthetic gene was amplified using the primers listed in Appendix A, carrying overhangs complementary to the termini of the linearized pET_Nm-fHbpDomA_MUC1. Four plasmids were generated, named pET-fHbp-DomA-MUC3x-L2_16_, pET-fHbp-DomA-MUC3x-4merA, pET-fHbp-DomA-MUC3x-4merB, and pET-fHbp-DomA-MUC3x-8merAB. The fusion constructs fHbp-DomA-L2_16_, fHbp-DomA-4merB, and fHbp-DomA-8merAB were finally transferred into plasmid pACYC [36]. To this aim, pACYC was linearized using the two primers PACYC_F and PACYC_R (Appendix A), and the fusion constructs were amplified from the pET plasmid using the primers listed in Appendix A. This procedure generated the three plasmids: pACYC-fhbp-DomA-MUC3x-L2_16_, pACYC-fhbp-DomA-MUC3x-4merB, and pACYC-fhbp-DomA-MUC3x-8merAB. The sequence of each construct was verified by DNA sequencing, using primers T7-P, T7-T, PACYC-up1, and PACYC-down1 (Appendix A). Finally, the plasmids were used to transform *E. coli* BL21(DE3)Δ60 obtaining the following recombinant strains: *E. coli* BL21(DE3)Δ60(pACYC-fHbp-DomA-MUC3x-L2_16_), *E. coli* BL21(DE3)Δ60(pET-fHbp-DomA-MUC3x-4merA), *E. coli* BL21(DE3)Δ60(pACYC-fHbp-DomA-MUC3x-4merB), and *E. coli* BL21(DE3)Δ60(pACYC-fHbp-DomA-MUC3x-8merAB).

### 2.3. OMV Preparation and Purification

Recombinant strains were grown in 600 mL LB at 30 °C under shaking at 200 rpm. At OD_600_ = 0.8, the recombinant antigen expression was induced by adding isopropyl-β-D-thiogalactoside (IPTG) at a final concentration of 0.1 mM. Normally, after 2–3 h, depending on the growth curve, the bacterial biomass was separated from the supernatant by centrifugation of the bacterial culture at 6000× *g* for 30 min. The supernatant was collected and filtered through a 0.22 μm pore size filter (Fisher Scientific part of ThermoFisher, Waltham, MA, USA) followed by the addition of 1 U/mL of benzonase (Sigma-Aldrich, St. Louis, MO, USA). For small-volume cultures, the supernatant was concentrated with a 100 kDa ultrafiltration membrane (Sigma-Aldrich, St. Louis, MO, USA), and the OMVs were collected by ultracentrifugation (200,000× *g* for 2 h). For large-volume cultures, the OMVs were purified from the supernatant through an ÄKTA flux Tangential Flow Filtration (TFF) (GE Healthcare, Chicago, IL, USA) using a 500 kDa Hollow Fibre cartridge UFP-500-C-3MA (GE Healthcare, Chicago, IL, USA), dialyzed against 1.5 L of sterile PBS and concentrated until a final volume of about 15 mL. Purified OMVs were filtered through a 0.22 μm pore size filter (Fisher Scientific part of ThermoFisher, Waltham, MA, USA) and the protein content was determined using DC Protein Assay (Bio-Rad, Hercules, CA, USA). The quality of the OMVs was analyzed using SDS-PAGE by loading 20 μg of total OMV proteins on a NuPAGE 4–12% Bis-Tris gel (Invitrogen, Waltham, MA, USA), which was finally stained with Coomassie Blue (Giotto, Sesto Fiorentino, Italy). The expression of each antigen was analyzed by performing a densitometric analysis using Image Studio Lite software v5.0 (LI-COR Biosciences, Lincoln, NE, USA).

### 2.4. Dynamic Light Scattering

The dynamic light scattering analysis allows the measurement of the size distribution profile of OMVs. The technique is based on the laser diffraction method using Zetasizer Nano-ZS90 (Malvern, UK). For measurement, the OMV batches were prepared at a final concentration of 0.5 mg/mL in PBS. The diameter of the OMVs was determined by measuring the 90° side scatter size at 25 °C. The final size was obtained from the average of three different measurements, each one obtained from 15 to 20 experimental runs.

### 2.5. Confocal Microscopy Analysis of Recombinant E. coli Strains

The surface localization of the L2 epitopes in recombinant strains was analyzed by confocal microscopy using anti-HPV16-L2 monoclonal antibodies K18 [28]. Briefly, the *E. coli* BL21(DE3)Δ60(pACYC-fHbp-DomA-MUC3x-L2_16_), *E. coli* BL21(DE3)Δ60(pET-fHbp-DomA-MUC3x-4merA), and *E. coli* BL21(DE3)Δ60(pACYC-fHbp-DomA-MUC3x-8merAB) recombinant strains were grown in LB at 30 °C. At OD_600_ = 0.5, 0.1 mM IPTG was added to the cultures, and bacteria were grown for two additional hours. Bacteria from 3 mL of each culture were harvested by centrifugation at 11,000× *g* for 2 min at 4 °C and re-suspended in 4% paraformaldehyde solution in PBS, incubated for 15 min at room temperature (RT), and then centrifuged at 7000× *g* for 2 min. Bacteria were washed three times with 1 mL PBS, suspended in 1 mL of blocking buffer (PBS containing 1% BSA), and incubated for 20 min at RT. Primary monoclonal antibodies against L2 of HPV 16 were diluted in PBS containing 1% BSA at a final concentration of 5 μg/mL and incubated for 1 h at RT. After two washes with PBS, bacteria were incubated for 20 min at RT with a solution containing the secondary Alexa Fluor^®^ 488-labelled goat anti-mouse antibody (Molecular Probes, Eugene, OR, USA) diluted 1:400 and the DAPI (ThermoFisher, Waltham, MA, USA) at the final concentration of 1:2500. Labeled bacteria were washed twice with PBS and placed on the slides (BioSigma, Cona (VE), Italy) with the ProLong Gold anti-fade reagent (Thermo Scientific part of ThermoFisher, Waltham, MA, USA). Confocal microscopy analysis was performed with an SP5 microscope (Leica, Wetzlar, Germany) and images were obtained using Leica LASAF4.0 software (Leica, Wetzlar, Germany).

### 2.6. Analysis of Surface Localization of L2 Epitopes Using Proteinase K Assay

Newly prepared OMV samples (2 μg) were treated with 100 μg/mL of Proteinase K (Applichem, Monza, Italy) in the presence or absence of 1% SDS. After incubation for 30 min at 37 °C, the peptidase inhibitor phenylmethylsulfonyl fluoride (PMSF) (Sigma-Aldrich, St. Louis, MO, USA) was added at a final concentration of 3 mM. The OMV proteins (0.05 to 0.5 μg) were separated using SDS-PAGE and the integrity of the DomA-L2 fusions was analyzed with Western Blot. Briefly, after separation on SDS-polyacrylamide gels, the proteins were transferred to PVDF membranes (Invitrogen, Waltham, MA, USA). The membranes were then incubated for 1 h at RT in 10% skimmed milk (Sigma-Aldrich, St. Louis, MO, USA), and 0.05% Tween in PBS under mild agitation. Subsequently, the membranes were incubated for 1 h at RT in a PBS solution containing 1% skimmed milk, 0.05% Tween, and 0.5 μg/mL of an anti-HPV16-L2 monoclonal antibody K18 [28]. After three washing steps in PBS containing 0.05% Tween, the filters were incubated for 1 h in a 1:4000 dilution of peroxidase-conjugated anti-mouse immunoglobulins (Sigma-Aldrich, St. Louis, MO, USA) in PBS containing 1% skimmed milk and 0.05% Tween. The membranes were washed three times with PBS and then the immunoreactive signals were detected with the ImageQuant LAS4000 (GE, Chicago, IL, USA) using the SuperSignal West Pico chemiluminescent substrate (Thermo Scientific part of ThermoFisher, Waltham, MA, USA).

### 2.7. Negative Staining Electron Microscopy Analysis

Empty OMVs, 4merA-OMVs, 4merB-OMVs, and 8merAB-OMVs were loaded onto a Q150R S (Quorum, Laughton, UK) copper 200-square mesh grid at a final dilution of 80 ng/μL in a saline buffer in a volume of 5 μL for each sample. After 1 min, the solution in excess was discarded using Whatman filter Paper No. 1. The grid was negatively stained using the NanoW (Nanoprobes, Yaphank, NY, USA) through an incubation of 30 s at RT and subsequently blotted with Whatman filter Paper No. 1. Finally, the negatively stained samples were allowed to air dry. A G2 Spirit Transmission Electron Microscope (Tecnai, Hillsboro, OR, USA) equipped with a CCD 2kx4k camera was used for sample acquisitions and micrographs were collected at a final magnification of 120,000×.

### 2.8. Interleukin 6 (IL-6) Reactogenicity Assay

THP-1 human leukemic monocyte cells were cultured in RPMI (Sigma-Aldrich, St. Louis, MO, USA) supplemented with 10% FBS. For the differentiation of the monocytes into macrophages, 10 ng/mL of phorbol 12-myristate 13-acetate (PMA) was added into the culture medium and maintained for 48 h. Then, after medium replacement with complete RPMI, cells were maintained in culture for an additional 24 h. Different amounts (diluted in a final volume of 100 μL RPMI) of “Empty” OMVs, 4merB-OMVs, or commercially available Bexsero vaccine (based on the OMV concentration per dose of vaccine) were added to 1.5 × 10^5^ differentiated macrophages. The OMVs were diluted with 10-fold serial dilution steps starting from a concentration of 1000 ng/mL, and plates were incubated ON at 37 °C. The amount of IL-6 released in supernatants was measured using the Human IL-6 Uncoated ELISA™ Kit (Thermo Fisher Scientific, Waltham, MA USA) following the manufacturer’s protocol. Briefly, Corning Costar ELISA plates were coated with anti-human IL-6 antibody (100 μL/well) by overnight incubation at 4 °C. The day after, the blocking solution was added to each well (Thermo Fisher Scientific, Waltham, MA USA) and subsequently, 100 μL/well of cell supernatants were transferred to the plates and incubated for 2 h at RT. Finally, biotin-conjugated anti-IL-6 human antibodies, streptavidin-HRP, and tetramethylbenzidine (TMB) substrates were added to each well according to the manufacturer’s instructions and the plates were read at 450 nm using a Varioskan apparatus (ThermoFisher, Waltham, MA, USA). The results were analyzed by comparing the value to the standard curve obtained with different concentrations of purified human IL-6 (2 to 200 pg/mL).

### 2.9. Animal Experiments

Four groups of 6–8 weeks old CD1 female mice (4 or 5 mice per group, Charles River Laboratories Italia, Lodi, Italy) were i.p. immunized (200 μL/mouse) with 10 μg/dose of L2_16_-OMVs, 4merA-OMVs, 4merB-OMVs, or 8merAB-OMVs, respectively, formulated with 2 mg/mL aluminum hydroxide (Alum) (InvivoGen, Toulouse, France). The vaccination with 8merAB-OMVs was also repeated in another group of four mice using 25 μg/dose of OMVs, formulated with 2 mg/mL Alum. For each experiment, three immunizations were performed at two-week intervals. One week after the last immunization, mice were sacrificed, and sera were collected from each mouse.

### 2.10. Enzyme-Linked Immunosorbent Assay (ELISA)

Sera from immunized mice were collected and analyzed with enzyme-linked immunosorbent assay (ELISA). Covalink 96-well plates (ThermoFisher, Waltham, MA, USA) were coated with 100 μL/well of 100 mM sodium carbonate solution containing 5 μg/mL of each synthetic peptide corresponding to L2 from HPV 16, 18, 31, 33, 35, 51, 59, and 6 (Genescript, Piscataway, NJ, USA). After an ON incubation at 4 °C, the plates were saturated with 1% BSA in PBS for 1 h at 37 °C. Mice sera were 3-fold serially diluted starting from 1:100 to 1:218,700 in a solution containing 0.1% BSA in PBS and incubated for 1 h at 37 °C. After three washes with 200 μL/well of 0.05% Tween in PBS, goat anti-mouse IgGs (total IgG, IgG1, and IgG2a) conjugated with alkaline phosphatase (Sigma-Aldrich, St. Louis, MO, USA) was added at a 1:2000 dilution. After 45 min at 37 °C and three washes with 0.05% Tween in PBS the substrate p-nitrophenyl phosphate (pNPP, Sigma-Aldrich, St. Louis, MO, USA) containing 100 mM glycine, 1 mM ZnCl_2_, 1 mM MgCl_2_ (100 μL/well) was added and the plates were read at 405 nm using the Varioskan apparatus (ThermoFisher, Waltham, MA, USA).

### 2.11. Eukaryotic Cell Cultures

HeLaT-K4 cells were cultivated in Dublecco modified Eagle medium (DMEM) (Sigma-Aldrich, St. Louis, USA) containing 1000 mg glucose, supplemented with 10% fetal bovine serum (Sigma-Aldrich, St. Louis, MO, USA), 1% glutamine (Sigma-Aldrich, St. Louis, MO, USA), 1% Pen/Strep (Sigma-Aldrich, St. Louis, MO, USA). Cells were cultured at 37 °C, 5% CO_2,_ and 95% humidity.

### 2.12. Pseudovirions Preparation

Different types of pseudovirions (PsVs) were prepared by cotransfection of the human fibroblast cell line 293TT with plasmids carrying humanized HPV L1 and L2 coding sequences and a reporter plasmid expressing Gaussia luciferase. Purification was performed by iodixanol gradient ultracentrifugation according to a previously described protocol [37].

### 2.13. Neutralization Assays

Sera were collected from immunized mice and diluted (1:50 to 1:12,150) in a 96-well tissue culture plate (Corning, New York, NY, USA). The sera were incubated with pseudovirus for 20 min, and after the addition of HelaT-K4 cells (3.3 × 10^5^ cell/mL), the plates were incubated under 5% CO_2_ at 37 °C for 48 h. After the incubation, 10 μL of the supernatant from each well was transferred to a 96-well white plate, and 100 μL/well of Gaussia luciferase substrate (PJK, Kleinblittersdorf, Germany) was added and luminescence was measured after 15 min. EC_50_ is defined as the titer of serum that could neutralize half of the pseudovirus.

## 3. Results

### 3.1. L2 Epitope Is Efficiently Expressed on the Surface of E. coli and Accumulated in OMVs

We previously showed that the lipoprotein factor H binding protein of *Neisseria meningitidis* (fHbp) [13]. In agreement with what was previously published [38], the protein protrudes out of the outer membrane even in the absence of the fHbp-specific “transporter” Slam [39]. Moreover, we reported that the N-terminal domain of fHbp (fHbp-DomA) could be efficiently exploited as a carrier to deliver polypeptides to the *E. coli* cell surface and that the fHbp-DomA-polypeptide fusions accumulated in the OMVs with high efficiency [13].

Therefore, we investigated whether fHbp-DomA could be utilized to deliver the L2 epitopes to the OMV compartment. In particular, we examined the potential of fHbp-DomA-MUC1 fusion as a carrier [13]. This fusion, which carries three copies of the MUC1 repeats at its C-terminus, was selected for its high level of expression in the vesicular compartment to favor structural flexibility at the junction between the DomA and the L2 epitope(s).

To assess the feasibility of the approach, we first test the expression of a single L2 epitope, from HPV16, as a fusion protein with DomA and MUC1 (see Section 2). As shown in Figure 1C, the fHbpDomA-MUC3x-L2_16_ fusion (hereinafter DomA-L2_16_) was expressed in the OMVs with high efficiency, representing approximately 24% of total OMV proteins as confirmed using densitometric scanning of the SDS-polyacrylamide gel. Moreover, the fusion protein was exposed on the surface of the OMVs, as determined using the proteinase K “shaving” assay (Figure 1D). In this assay, purified OMVs were treated with proteinase K and subsequently, the integrity of the fusion protein was analyzed by Western blot, using a monoclonal antibody specific for the L2 epitope. As shown in Figure 1D, the band corresponding to DomA-L2_16_ almost completely disappeared after protease treatment.

Upon demonstrating that the L2 epitope is efficiently transported to the OMV compartment, we next investigated whether strings of L2 epitopes from different HPV serotypes could similarly be incorporated in the vesicles. Three different DomA fusions were generated: DomA-4merA carrying the L2 epitopes from HPV serotypes 16, 31, 51, and 6; DomA-4merB carrying the L2 epitopes from serotypes 18, 33, 35, and 59; and DomA-8merAB, in which 4merA and 4merB were fused. In all three constructs, each L2 epitope was separated from the other by a Gly-Gly-Pro (GGP) spacer. The plasmids carrying the synthetic genes coding for the three fusions (Appendix A) were used to transform *E. coli* BL21(DE3)Δ60 generating the three recombinant strains *E. coli* BL21(DE3)Δ60(pET-fHbp-DomA-4merA), *E. coli* BL21(DE3)Δ60(pACYC-fHbp-DomA-4merB), and *E. coli* BL21(DE3)Δ60(pACYC-fHbp-DomA-8merAB). The OMVs purified from the culture supernatants of the three strains were analyzed with SDS-PAGE. As shown in Figure 1C, similarly to DomA-L2_16_, all three fusions were efficiently incorporated in the OMVs (from 13% to 16% of total OMV proteins). Moreover, the fusion proteins, and in particular the strings of the L2 epitopes, were well exposed on the surface of the outer membrane as confirmed using the proteinase K “shaving” assay on engineered OMVs (Figure 1D). The surface exposure of the epitopes was also demonstrated with confocal microscopy of the engineered bacteria stained with the monoclonal antibody K18, specific for the L2 epitope of HPV16 [40], (Figure 1E). Finally, electron microscopy analysis showed that 4merA-OMVs, 4merB-OMVs, and 8merAB-OMVs had similar morphology and size (Figure 1F).

### 3.2. Immunization with L2-Engineered OMVs Elicits L2-Specific IgG Titers

Next, we investigated whether the OMVs engineered with the L2 epitopes could induce L2-epitope-specific antibodies. To achieve this aim, three groups of five CD1 mice each were immunized three times, two weeks apart, with 10 μg of either 4merA-OMVs, 4merB-OMVs, or 8merAB-OMVs. Seven days after the last immunization, sera were collected and used to measure the L2-specific IgG titers by ELISA. The assay was carried out by coating the plates with each of the eight synthetic peptides corresponding to the selected L2 epitopes. From the ELISA titers reported in Figure 2, the following conclusions can be drawn. First, 4merA-OMVs and 4merB-OMVs immunizations elicited antibodies, which recognized their own L2 epitopes. Second, the sera from mice immunized with 4merA-OMVs and 4merB-OMVs also cross-reacted with the epitopes present in the other construct, except for the L2 of serotype 59, which was poorly recognized by the anti-4merA-OMV sera. Cross-recognition was particularly effective in sera from mice immunized with 4merB-OMVs, which produced titers > 1 × 10^3^ against all eight synthetic peptides in most of the vaccinated mice. Third, 8merAB-OMVs induced antibodies specific for all L2 epitopes. However, when 10 μg was used, the titers against serotypes HPV16, 6, 18, 33, and 35 appeared to be slightly lower than the titers obtained with 4merB-OMVs. This was probably due to the lower amount of each epitope present in the 10 μg of 8merAB-OMVs with respect to 10 μg of the tetramer construct. Indeed, when the immunization was repeated using 25 μg of 8merAB-OMVs, the titers reached the highest levels against all eight L2 peptides. Finally, the inspection of individual mouse serum (Figure 2B) revealed that titer variability was not epitope-specific.

The above conclusions have to be taken with a certain degree of caution since the number of animals per group was limited (4 mice). Moreover, such conclusions do not take into consideration the possible difference in the expression level of the three constructs in the OMVs—an expression that was not rigorously quantified.

We finally investigated the IgG isotypes as a way to establish whether the OMVs engineered with the L2 epitopes elicited a Th1-skewed immune response. In particular, the IgG1 and IgG2a titers were measured in the pool of sera from mice vaccinated with 4merB-OMVs—our most promising vaccine candidate as determined by total IgGs and neutralization titers. As shown in Appendix A, in line with what was previously observed [14], high titers of IgG2a antibodies were detected.

In conclusion, OMVs carrying L2 epitopes were capable of eliciting high levels of anti-L2 antibodies. Moreover, the immunization with 10 μg of 4merB-OMVs was particularly effective—the formulation did not only elicit antibodies that cross-reacted with all eight L2 serotypes tested, but also performed equally well concerning the immunization with 25 μg of 8merAB-OMVs.

### 3.3. L2-Specific IgGs Elicited by OMV Immunization Neutralize HPV In Vitro

We next asked the question of whether the L2-specific antibodies induced by OMV immunization had functional activity. Since no reliable animal models are available to study human HPV infection, the in vitro pseudovirus neutralization assay (PBNA) was used [31]. In this assay, pseudovirus composed of L1 and L2 from selected HPV serotypes and Gaussia luciferase were used to infect HeLaT-K4 cells in the presence or absence of mouse sera, and the capacity of serum antibodies to inhibit pseudovirus infection was quantified following the luciferase activity in the cells.

Pseudoviruses corresponding to all eight HPV serotypes selected for the L2 polytope constructs were prepared and systematically tested in the in vitro neutralization assay using the sera from mice immunized with 4merA-OMVs, 4merB-OMVs, and with the 10 μg/dose of 8merAB-OMVs.

As shown in Figure 3, the three vaccine formulations elicited neutralization titers against all serotypes tested. The two tetramers induced antibodies that not only neutralized the corresponding pseudoviruses but also the pseudoviruses belonging to the other four serotypes. However, in line with the ELISA titers, the neutralization titers elicited by the 4merA-OMVs construct were low against the HPV59 serotype. Furthermore, HPV51 pseudovirus appeared to be the most difficult pseudovirus to neutralize, regardless of the formulation used. Paralleling the ELISA titers, the 4merB-OMVs outperformed both the 4merA-OMVs and the 8merAB-formulation given at 10 μg/dose. This could be appreciated by comparing the graphs shown in Figure 3B; the neutralization titers elicited by the 4merB-OMVs against each serotype were never lower, and in most of the cases, higher than the titers induced by the other two formulations. Again, variability in fusion expression levels, which were not rigorously determined, could at least partially explain the difference among the neutralization titers of the three OMV vaccines.

The availability of both ELISA and neutralization titers from each mouse serum allowed the evaluation of whether there was a correlation between the titers of the epitope-binding antibodies and functional (neutralizing) antibody titers. In general, sera with the highest ELISA titers against a specific serotype performed well in the neutralization assay and vice versa (Appendix A).

### 3.4. Set-Up of a Laboratory Scale Production Process of 4merB-OMVs

Data reported above indicated that the 4merB-OMVs is an attractive HPV vaccine candidate. The vaccine elicits broadly protective neutralizing antibodies at quantities (10 μg/dose) that are expected to be well tolerated in humans, considering the excellent safety profile of the Bexsero vaccine, which contains 25 μg of OMVs + 3 mg/mL of Alum.

In view of future clinical studies, we investigated the possibility of setting up a reproducible production process for our 4merB-OMVs vaccine. As shown in Figure 4, the OMV production process appeared to be highly reproducible in terms of kinetics of bacterial growth (Figure 4A), protein composition (as determined using 1D electrophoresis (Figure 4B)), OMV yield that was 11.4 ± 1.05 mg/L (Figure 4C), and size distribution (diameter = 40.2 ± 3.4) (Figure 4D). Finally, the 4merB-OMV vaccine was tested with the IL-6 release assay—a validated assay used to follow the reactogenicity and lot consistency of the Bexsero vaccine [40]. Increasing quantities of 4merB-OMVs were added to differentiated THP-1 cells and the IL-6 released in the supernatant was compared to the amount of IL-6 released by different quantities of commercially available Bexsero vaccine. As shown in Figure 4E, the 4merB-OMVs and Bexsero released similar quantities of IL-6.

Overall, these data support the feasibility of a pilot-scale production process for future toxicity studies in animals and Phase I studies in human volunteers.

## 4. Discussion

Current HPV vaccines, based on the major L1 capsid protein, are highly effective in preventing HPV infections. However, vaccine costs and serotype specificity, which requires the combination of more than one VLP lots, are major hurdles that limit the introduction of global vaccination campaigns that will drastically reduce HPV-related pathologies—cancer above all.

In this work, we presented a new formulation that has the potential to overcome the limitations of the L1-based vaccines. Taking advantage of previous work showing that the L2 HPV protein carries a conserved neutralizing B cell epitope, and that a string of L2 epitopes selected among eight relevant HPV serotypes provides broad cross-protection [34], we investigated whether *E. coli* OMVs could be decorated with L2 repeats and whether such OMVs could elicit anti-HPV functional antibodies. Our data showed that, indeed, the expression of the L2 repeats fused to the N-terminal domain of neisserial fHbp was highly efficient; the fusion proteins represented approximately 13–16% of total OMV proteins. Moreover and importantly, the immunization of mice with a single preparation of engineered OMVs elicited high titers of anti-L2 antibodies, which could efficiently neutralize the in vitro infection by a panel of eight HPV serotypes, as determined with the pseudovirus-based neutralization assay.

Interestingly, based on the sequence homology, Spagnoli et al. [41] predicted that the L2 epitopes from the serotypes 16, 31, 51, 6, 18, 33, 35, and 59 were necessary and sufficient to design vaccines that could protect against more than 90% of all circulating oncogenic HPV and genital wart associated strains, including the serotypes which are the target of the current nonavalent Gardasil vaccine [34]. Our neutralization data showed that the expression of the 18-33-35-59 tetramer in OMVs (4merB) was sufficient to neutralize the homologous pseudovirus and the 16, 31, 51, and 6 pseudoviruses with high efficiency. Although the vaccine neutralization capacity has not been tested against other serotypes, our results suggest that the L2 epitopes are presented on the surface of the OMVs in a configuration that allows the elicitation of antibodies with a cross-protective activity broader than what was originally predicted.

A relevant question is whether the neutralizing activity induced by our L2-expressing OMVs would be sufficient to protect against HPV infection in humans and how the neutralization titers compare with the titers induced by the L1-based vaccines. Although head-to-head comparisons with commercially available vaccines in mice could be performed, the real evidence can only be obtained in humans. Romanowski et al. [42] reported that in humans, the HPV16 neutralizing antibody titers induced by the HPV-16/18 AS04-adjuvanted vaccine and measured 7 months after the third vaccination dose were in the range of 2 × 10^4^. Should our OMV vaccine be in humans as immunogenic as in mice, the neutralization titers of the L2-based vaccine and the L1-based vaccine would not be too dissimilar. Nevertheless, the threshold of anti-L2 antibody titers for protection against HPV infection in humans is not known.

Clinical trials using L2-based vaccines are considered particularly complicated in view of the availability of effective anti-L1-based vaccines. However, the human population appears to be seronegative with respect to the L2 N-terminal epitope. Neither natural infection nor vaccination have been reported to elicit antibodies against this particular L2 epitope [34]. Therefore, it is possible to envisage studies, whereby volunteers vaccinated with an L1 vaccine receive our OMV (4merB) vaccine, and vaccine immunogenicity and duration of anti-L2 responses are evaluated by comparing anti-L1 and anti-L2 antibody titers for control volunteers. Subsequently, OMV vaccine efficacy can be established following the incidence of infection/disease caused by non-L1 vaccine strains.

A second possible clinical application of our OMV vaccine would be its combination with one of the existing L1 vaccines. We believe that such a combination would be extremely attractive for three main reasons. First, the anti-L1 and anti-L2 antibodies are expected to synergize, thus providing extremely high neutralization titers. Second, the neutralizing antibody titers should be further enhanced by the potent adjuvant contribution of the OMVs. Third, considering the broad protective activity of the string of L2 epitopes, the vaccine combination is likely to become a universal PAN-HPV vaccine. Nonetheless, the limitation of this vaccine formulation would be to overcome the problems of costs and stability of the existing vaccines, and therefore the use of such PAN-HPV vaccine risks being restricted to high-income countries.

The indisputable advantages of OMV-based vaccines are the simplicity of the production process, which could be easily set up in any local production facility at low costs. Indeed, our experiments showed that, at least under laboratory conditions, the production process of our OMV (4merB) vaccine is robust and reliable. Considering the use of a 2 L fermentation unit, using our “proteome minimized” strain [12], we routinely obtain 20–50 mg of OMVs/L of culture and assuming a vaccination schedule of three injections of 10 to 20 μg of OMVs/dose, up to 10^5^ three-dose vaccines could be produced from a small 100 L fermentor.

To deliver the strings of the L2 epitopes to the surface of the OMVs, we used the 120-amino-acids-long N-terminal domain of the neisserial fHbp carrying three copies of the 20 amino acid MUC1 epitope at its C-terminus. The motivation to use this fusion is threefold. First, we previously demonstrated that such fusion was exposed on the surface of the outer membrane and accumulated in the vesicular compartment with remarkable efficiency, and the fusion represented more than 20% of total OMV proteins [13]. Second, the MUC1 repeat is expected to provide flexibility to the L2 polyepitope, thus facilitating the elicitation of proper humoral responses. Third, MUC1 is an interesting tumor-specific epitope; the epitope with the sequence GVTSAPDTRPAPGSTAPPAH was found repeated 20 to 150 times in the extracellular domain of the transmembrane glycoprotein Mucin1 (MUC1) and constitutes the so-called “variable number of tandem repeats region (VNTR)” [43]. In normal epithelia, VNTR is highly glycosylated in Serine and Threonine, whereas in most adenocarcinomas such as those of breast, ovary, colon, pancreas, lung, head, and neck, as well as in premalignant lesions, MUC1 becomes over-expressed and hypoglycosylated. Several vaccines based on VNTR have been tested in Phase I/II studies and such vaccines have also been proposed for the prevention of different adenocarcinomas [44]. We have recently replaced MUC1 with other hinge regions, with no homologies to human proteins, and these fusions were expressed with similar efficiencies in the OMVs. However, considering that the MUC1-based vaccines have shown an excellent safety profile in humans [45], we think that the HPV/MUC1 combination could be an interesting candidate for the prevention of HPV-associated tumors.

## Figures and Tables

**Figure 1 vaccines-11-01582-f001:**
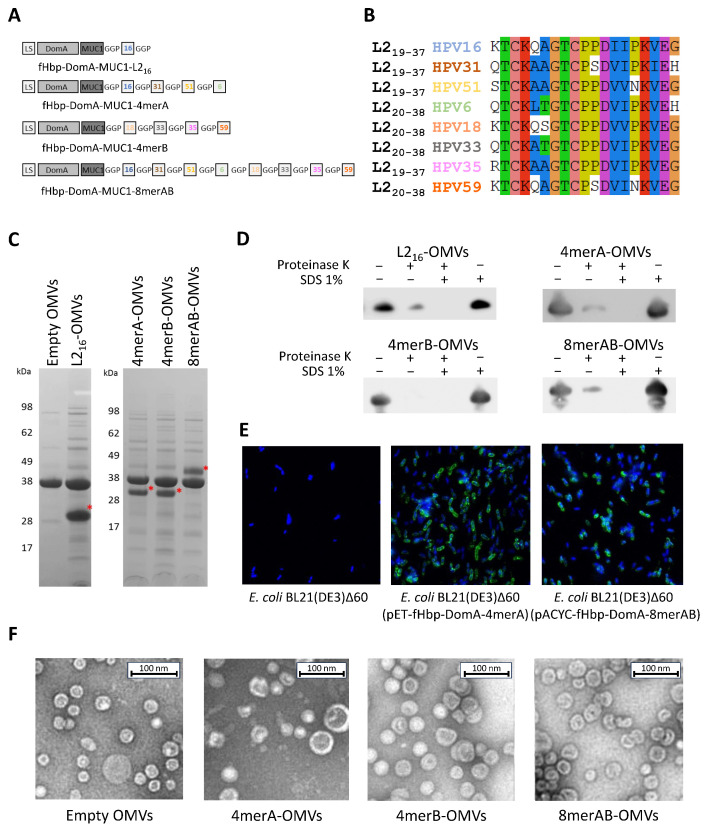
Engineering of OMVs with L2 epitope. (**A**) Schematic representation of the fusion proteins carrying the L2 epitope(s) at the C-terminus. LS: leader sequence; DomA: N-terminal domain of neisserial factor H binding protein; MUC: human Muc1 repeat; GGP: Glycine-Glycine-Proline spacer; and boxed numbers indicate the L2 epitope of the corresponding HPV types. (**B**) Amino acid alignment of the eight L2 neutralizing epitopes of the HPV serotypes selected for OMV engineering. (**C**) SDS-PAGE analysis of engineered OMVs. Aliquots of 20 μg of OMVs purified from the culture supernatants of the engineered *E. coli* BL21(DE3)Δ60 strains were loaded on a polyacrylamide gel and stained with Coomassie Blue. As a negative control, the “Empty OMVs”, purified from *E. coli* BL21(DE3)Δ60 strain, were loaded. The red asterisk (*) indicates the band of interest. (**D**) Assessment of L2 epitope localization on engineered OMVs. Purified OMVs (1 μg) were treated for 30 min with proteinase K in the presence (+) or absence (−) of SDS and the integrity of the fusion proteins was analyzed by Western blot using an anti-L2_16_ monoclonal antibody (see Text for details). (**E**) Assessment of localization of L2 epitope by confocal microscopy. Bacterial cells of *E. coli* BL21(DE3)Δ60, *E. coli* BL21(DE3)Δ60(pET-fHbp-DomA-MUC3x-4merA), and *E. coli* BL21(DE3)Δ60(pACYC-fHbp-DomA-MUC3x-8merAB) were stained with a mAb specific for the L2 epitope of HPV16 (see Materials and Methods) (antibody binding: green; and nucleus: blue). (**F**) Negative staining of OMVs, visualized using electron microscopy. Five μL of OMVs purified from *E. coli* BL21(DE3)Δ60, *E. coli* BL21(DE3)Δ60(pET-fHbp-DomA-MUC3x-4merA), *E. coli* BL21(DE3)Δ60(pET-fHbp-DomA-MUC3x-4merB), and *E. coli* BL21(DE3)Δ60(pACYC-fHbp-DomA-MUC3x-8merAB) were negatively stained with NanoW for 30 s and micrographs were acquired using a G2 Spirit Transmission Electron Microscope (Tecnai, Hillsboro, OR, USA), at a final magnification of 120,000×.

**Figure 2 vaccines-11-01582-f002:**
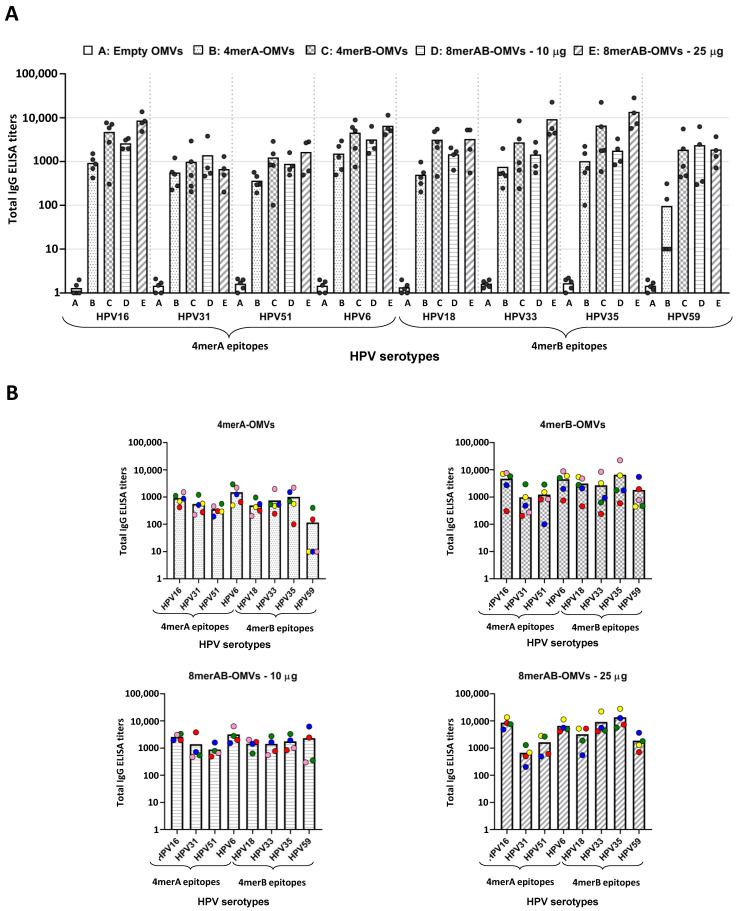
Anti-L2 ELISA titers in sera from mice immunized with engineered OMVs. A group of four or five CD1 mice was immunized (three immunizations) with “Empty OMVs” (negative control), 4merA-OMVs, 4merB-OMVs, 8merAB-OMVs (10 μg/dose), and 8merAB-OMVs (25 μg/dose) and sera were collected 7 days after the last immunization. Total IgG titers of each mouse serum against the different L2 epitopes were analyzed with ELISA using plates coated with the corresponding synthetic peptides (0.5 μg/well). (**A**) Cumulative representation of the titers elicited by sera from mice immunized with the different vaccine formulations tested against all selected HPV serotypes. Each vaccinated mouse is represented as a grey circle and ELISA titers correspond to the serum dilution that gives an OD_405_ value = 1.5 expressed on a logarithmic scale. (**B**) Representation of the ELISA titers against the eight L2 epitopes grouped by vaccine formulation. Each mouse serum is represented by a circle with a different color code. Graphs were generated using GraphPad Prism v8.0.2 software.

**Figure 3 vaccines-11-01582-f003:**
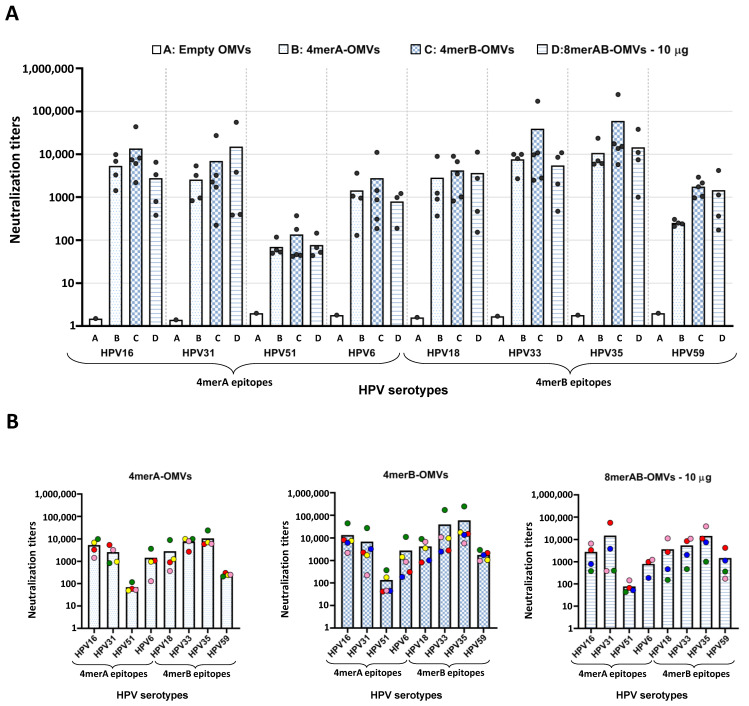
Neutralization titers of OMV-based vaccines. A group of four or five CD1 mice was i.p. immunized with “Empty OMVs” (negative control), 4merA-OMVs, 4merB-OMVs, or 8merAB-OMVs. Sera were collected 7 days after the last immunization and analyzed by pseudovirus-based neutralization assay (PBNA) using eight different HPV pseudoviruses (see Text for details). (**A**) Cumulative representation of the neutralization titers elicited by sera from mice immunized with the different vaccine formulations tested against all selected HPV serotypes. Each vaccinated mouse is represented as a grey circle apart from the negative control (“Empty” OMVs) in which the sera were pooled. In the graph, the EC_50_ value is represented, defined as the titer of serum that could neutralize half of the pseudovirus. (**B**) Representation of the neutralization titers against the eight pseudoviruses grouped by vaccine formulation. Each mouse serum is represented by a different color code. Graphs were generated using GraphPad Prism v8.0.2 software.

**Figure 4 vaccines-11-01582-f004:**
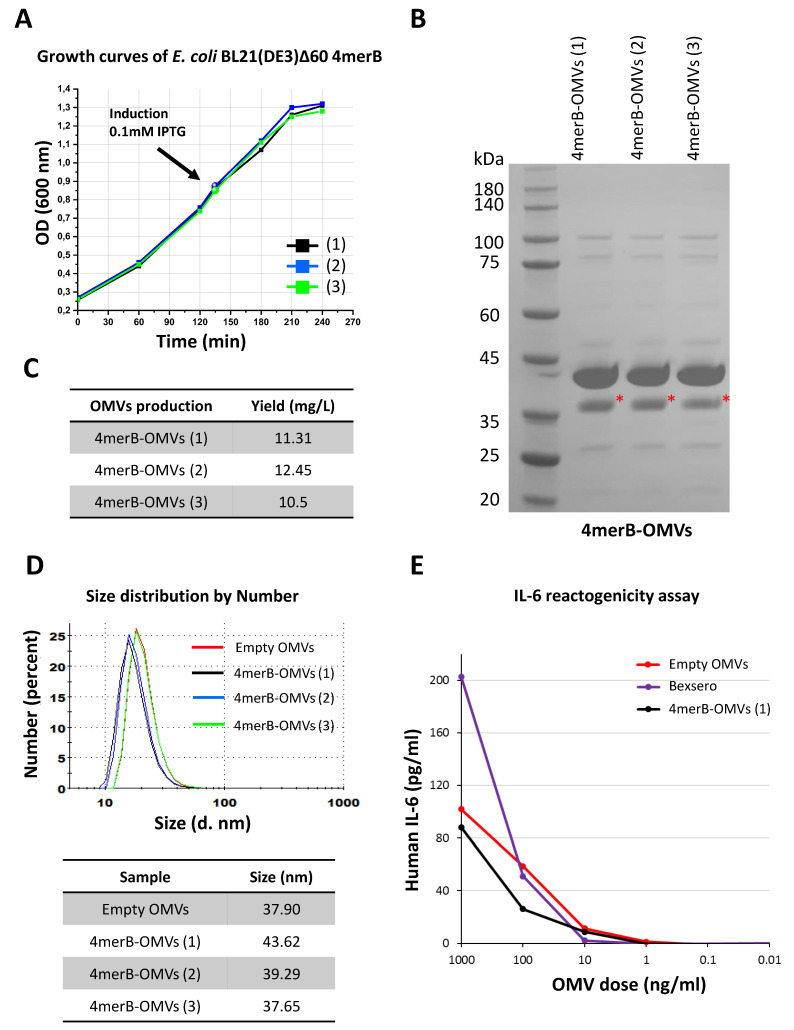
Reproducibility of the 4merB-OMVs laboratory scale production process. (**A**) *E. coli* BL21(DE3)Δ60(pACYC-fHbp-DomA-MUC3x-4merB) strain was grown in LB in triplicate (growth curves 1, 2, and 3) starting from three different overnight cultures and growth was monitored at 30-min intervals. When the cultures reached an OD_600_ value of 0.8, the expression of the DomA-L2 fusions was induced by the addition of 0.1 mM of IPTG. Three different OMV batches were purified from the bacterial culture supernatants through TFF. (**B**) Aliquots of 20 μg of OMVs purified from each culture were loaded on an SDS-PAGE polyacrylamide gel and stained with Coomassie Blue. The red asterisk indicates the band corresponding to DomA-4merB fusion. (**C**) Amount of 4merB-OMVs (expressed as mg/L of total proteins) recovered from each culture. (**D**) Size distribution profile of the three 4merB-OMVs preparations (black, blue, and green curves) determined with DLS using NanoZS90. The size of the 4merB-OMVs was compared with the purified “Empty” OMVs derived from recipient *E. coli* BL21(DE3)Δ60 (red curve). The table reports the means of the vesicle diameter (nm) of each 4merB-OMV batch and of “Empty” OMVs. (**E**) IL-6 release assay. THP-1 human leukemic monocyte cells (1.5 × 10^5^ cells in 100 μL/well) differentiated to macrophages with PMA were incubated with different amounts of either Empty OMVs, 4mer-OMVs, or OMVs present in the Bexsero vaccine in a final volume of 200 μL/well. The IL-6 released in the supernatant was measured in duplicate with ELISA.

## Data Availability

All data supporting the findings of this study are available within the paper and its Appendix A.

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
