# Peer review of "Bacterial Outer Membrane Vesicles as a Platform for the Development of a Broadly Protective Human Papillomavirus Vaccine Based on the Minor Capsid Protein L2"

_vaccines, 2023, doi:10.3390/vaccines11101582_

Round 1

Reviewer 1 Report

Tamburini et al developed a L2 epitope-based HPV vaccine using OMVs. They found ip immunization of the vaccine induced high level of serum IgG and a broadly neutralization antibody against the 8 types of HPV containing the vaccine.

Major comments:

1       Several neutralizing epitopes have identified on L2 N-terminal. Please describe why choice 20-38aa epitope, but not others? Do you have tested other epitopes, if have, please describe in discussion section.

2       Author confirmed the L2-epitops expressed on surface of E. Coli and OMV. Is it necessary and import for the immunity of the vaccine?

3       Why did authors use ip vaccination route, but not others, such as im, sc, or id route?

4       Since authors suggest that the vaccine induced broadly neutralizing activity, do you have data should IgG titer and neutralization Ab titer against other type epitopes and pseudotype HPVs?

Minor comment

1.     In Materials and Method section, author described using total IgG, IgG1 and IgG2a second antibodies for ELISA assay, but reviewer can not found IgG1 and IgG2a data.

2.     Reviewer do not sure whether it is appropriate to put Table 1 (Primers List) in text. If it is not so impact, the table can be moved to supplementary section.

Author Response

Reviewer 1

Tamburini et al developed a L2 epitope-based HPV vaccine using OMVs. They found ip immunization of the vaccine induced high level of serum IgG and a broadly neutralization antibody against the 8 types of HPV containing the vaccine.

 Major comments:

  • Several neutralizing epitopes have identified on L2 N-terminal. Please describe why choice 20-38aa epitope, but not others? Do you have tested other epitopes, if have, please describe in discussion section.

We have not tested other L2 epitopes. We selected the 20-38aa epitope since it has been extensively used to make an effective recombinant vaccine based on strings of L2 epitopes fused to thioredoxin (reference 27-33 in the manuscript). We believe this was sufficiently explained in the Introduction Section.      

2       Author confirmed the L2-epitops expressed on surface of E. Coli and OMV. Is it necessary and import for the immunity of the vaccine?

      This is a very good point. Although not described in the manuscript, we tested different carrier proteins for the expression of the L2 epitopes. Interestingly, the expression in the OMVs was efficient but the epitopes were poorly exposed on the surface, as judged by the proteinase K shaving assay. When we immunized mice, the anti-L2 antibody titers were low. Therefore, surface localization appears to be an important condition to elicit functional antibodies even though other factors, such as disulfide bonding, are expected to play a key role. Since we did not investigate the mechanisms determining the elicitation of functional Ab responses, we decided not to include these data in the manuscript.  

  • Why did authors use ip vaccination route, but not others, such as im, sc, or id route?

We used the ip route since is very simple and in general it is very effective in inducing humoral responses. We agree that ip vaccination is rarely used in humans. With other OMV vaccines we compared ip vs im or sc immunization routes and we obtained similar results (see for example Koenig et al. 2021, Ref 14).

  • Since authors suggest that the vaccine induced broadly neutralizing activity, do you have data should IgG titer and neutralization Ab titer against other type epitopes and pseudotype HPVs?

We do not have these data but using the same string of L2 epitopes (L2-thioredoxin fusion) broad cross-reaction was reported (please Spagnoli et al. 2017, Ref. 42 and Pouyanfard, S., et al 2018, Ref. 34).

Minor comment

  1. In Materials and Method section, author described using total IgG, IgG1 and IgG2a second antibodies for ELISA assay, but reviewer cannot found IgG1 and IgG2a data.

These data can be found in the Supplementary Material as Figure S3.

  1. Reviewer do not sure whether it is appropriate to put Table 1 (Primers List) in text. If it is not so impact, the table can be moved to supplementary section.

      We agree. Table 1 has been moved to the Supplementary material.

Reviewer 2 Report

Tamburini et al. described a preclinical HPV vaccine candidates based on bacterial outer membrane vesicles with a conserved, cross-reactive epitope from the L2 protein of HPV. L2-based vaccines have been proposed as alternatives for the L1-based vaccines, such as Gardasil and Cervarix, for their ability to induce a broad immune response and their potentially lower manufacturing costs. 

Major comments

·         Lines 75-77, Please, note that there are 6 approved HPV vaccines (Cervarix, Gardasil, Gradasil-9, Cecolin, Walrinvax and Cervavax). Therefore, the text should be amended: ‘Fortunately, a number of preventive HPV vaccines are available (Cervarix, Gardasil, Gradasil-9, Cecolin, Walrinvax and Cervavax). Of those, the bivalent HPV-16/18 Cervarix®, the quadrivalent HPV-6/11/16/18 vaccine Gardasil®, and Gardasil®9, a nonavalent HPV-6/11/16/18/31/33/45/52/58 vaccine, are most widely used. All of them are based on the L1 protein and take advantage of the fact that when expressed in yeast or insect different production cells L1 self-assembles and forms virus-like particles (VLPs) morphologically and antigenically highly similar to native virions26.

·         Line 82, the statement that the HPV vaccines are ‘are highly HPV 81 type-restricted with very limited cross-protection’ should be amended. Cervarix has been shown to give significant protection against several non-vaccine HPV types (https://www.ema.europa.eu/en/documents/product-information/cervarix-epar-product-information_en.pdf).

·         Figure 1B – check and correct numbering of L2 epitopes, in some cases the first amino acid is amino acid 19 (instead of 20). Why did the authors decide to include aa 20-38, while the introduction mentions that the cross-protective epitope cover amino acids 17-38?

·         Lines 351-353, Figure 1F does not seem to support the conclusion that the OMV’s ‘had similar sizes’. The particles seem to range from 10 to 30 or 40 nm. Overall, the 8mer-OMVs appear to be smaller than the 4mer-OMVs. The authors should show size distributions, similar to Figure 5D, for the different OMVs.

·         Lines 368-371, Please, mention that number of animals per group was too small to draw statistically significant conclusion. The apparent differences between 4mers and 8mer at 10µg could also be due to biological variability in one or mice, or assay variability. Is the possible explanation for the slightly lover immunogenicity of the 8mer AB-OMVs correct? Based on the sizes of approx. 34kDa and 43kDa for the 4mer and 8mer fusion proteins respectively, the difference in total amount of protein would be approx. 25%. However, the 8mer contains twice as many cross-reactive L2 epitopes. So, one could also argue that the 8mer should be more immunogenic. Have the authors quantified the number of copies of the 4mers and 8mers that are present of the particles, e.g., by quantifying MUC1?

·         Lines 410-420, Similar to the comment on lines 368-371, a higher number of fusion proteins on the 4merB-OMVs could also explain the neutralization data (Lines 410-420 and Figure 3), where 4merB, rather unexpectedly, outperforms 4merA-OMVs for the HPV types that are present in 4merA.

·         Figure 4, represents data from Figures 2 and 3 in a different manner. Figure 4 should be moved to the Supplementary data and the description/conclusion (lines 433-438) can be shortened: ‘In general sera with the highest ELISA  titers against a specific serotype also performed well in the neutralization assay and vice versa (Figure S…).

·         Lines 439-441, Delete sentences. In the absence of data, it is only an assumption that does not add value to the manuscript.

·         Lines 517-521, The prediction by Spagnoli et al. was based on sequence differences between HPV L2 epitopes, not on experimental data. They did not test different compositions in reference 41, they only compared HPV16 vs HPV-8x. Therefore, the conclusion that the OMVs might present the epitopes in a manner that leads to broader cross-reactivity (Lines 523-527) cannot be drawn. That would require a direct comparison between 4merB-OMVs and PfTrx-L2(18-33-35-39)-OVX313 nanoparticles.

·         Lines 532-539, The inference that the authors try to make, is too speculative. The mechanisms of action of L1- and L2-based vaccines are probably different and mouse data cannot reliably predict (neutralizing) antibody titers in humans. Furthermore, the durability of the anti-L2 antibodies after vaccination with the OMV particles in humans is unknown. Proposed amendment of the text: ‘A relevant question is whether the neutralizing activity induced by our L2-expressing OMVs would be sufficient to protect HPV infection in humans and how the neutralization titers compare with the titers induced by the L1-based vaccines. Although head-to-head comparisons with commercially available vaccines in mice could be performed, the real evidence can only be obtained in humans. This said, the threshold of anti-L2 antibody titers for protection against HPV infection inn humans is not known.  

·         Lines 553-559, The authors suggestion to combine with existing L1-based vaccines contradicts their earlier statement that the L1-based vaccines are too expensive and dependent on the cold chain for transport and distribution. The authors should point out this contradiction or perhaps suggest to combine with E.coli based L1 vaccines.

·         Lines 585-587, The authors conclude that the combination with a tumor-specific epitope from MUC1 generates an interesting vaccine candidate to prevent HPV-associated tumors and they refer to the safety profile in a colon cancer study. However, is anything known about the safety of inducing anti-MUC1 antibodies in healthy people, the main target population of a preventative HPV vaccine?

Minor comments

·         Line 37, delete ‘amazing’

·         Line 45-47, Suggestion to change to: ‘To develop cost-effective vaccines against infectious diseases and cancer, our laboratories have been working on a new vaccine platform based on engineered OMVs from non-pathogenic Escherichia coli (E. coli) derivatives over the last few years.

·         Table 1, move to supplementary materials.

·         Lines 281-287, move to introduction.

·         Lines 293-301, partly repeating the description in Materials and Methods. Can be shortened substantially: ‘To test the feasibility of the approach, we first test expression of a single 293 L2 epitope, from HPV16, as a fusion protein with DomA and MUC1 (hereinafter DomA-L216) as described in Materials and Methods. As shown in Figure 1C, DomA-L216 was expressed in the OMVs etc….’.

·         Line 302, delete ‘the’ in ‘…approximately the 24%’ of total OMV proteins…’.

·         Line 303, ‘Moreover, the fusion protein was exposed….’.

·         Lines 322-326, refer to Materials and Methods instead of repeating the method. Note that the legend mentions 2% paraformaldehyde, while 4% is mentioned in the methods.

·         Lines 400-401, A vaginal challenge model has been used to study HPV vaccine efficacy in mice. The authors should explain why they chose the in vitro neutralization assay.

·         Line 402, ‘According to In this assay,….’

·         Line 406, Production of the pseudoviruses should be described in Materials and Methods (or a reference/source should be added).

·         Lines 460-471, Delete sentences (already described in Materials and Methods).

Author Response

Reviewer 2

Tamburini et al. described a preclinical HPV vaccine candidates based on bacterial outer membrane vesicles with a conserved, cross-reactive epitope from the L2 protein of HPV. L2-based vaccines have been proposed as alternatives for the L1-based vaccines, such as Gardasil and Cervarix, for their ability to induce a broad immune response and their potentially lower manufacturing costs. 

As a general comment, we wish to thank this reviewer for his/her very competent reviewing. We have incorporated all suggestions/recommendations and we believe that the manuscript has been substantially improved.

Major comments

  • Lines 75-77, Please, note that there are 6 approved HPV vaccines (Cervarix, Gardasil, Gradasil-9, Cecolin, Walrinvax and Cervavax). Therefore, the text should be amended: ‘Fortunately, a number of preventive HPV vaccines are available (Cervarix, Gardasil, Gradasil-9, Cecolin, Walrinvax and Cervavax). Of those, the bivalent HPV-16/18 Cervarix®, the quadrivalent HPV-6/11/16/18 vaccine Gardasil®, and Gardasil®9, a nonavalent HPV-6/11/16/18/31/33/45/52/58 vaccine, are most widely used. All of them are based on the L1 protein and take advantage of the fact that when expressed in yeast or insect different production cells L1 self-assembles and forms virus-like particles (VLPs) morphologically and antigenically highly similar to native virions26.

Thank you so much for pointing this out. We confess that we have erroneously neglected the other three vaccines that were developed later and, as pointed out by this reviewer, are not extensively used in Western Countries. We have “copied and pasted” the sentence suggested by this reviewer. 

  • Line 82, the statement that the HPV vaccines are ‘are highly HPV 81 type-restricted with very limited cross-protection’ should be amended. Cervarix has been shown to give significant protection against several non-vaccine HPV types (https://www.ema.europa.eu/en/documents/product-information/cervarix-epar-product-information_en.pdf).

This is correct. We have modified the sentence, which now reads “in general they are HPV type-restricted with a limited cross-protection”.

  • Figure 1B – check and correct numbering of L2 epitopes, in some cases the first amino acid is amino acid 19 (instead of 20). Why did the authors decide to include aa 20-38, while the introduction mentions that the cross-protective epitope cover amino acids 17-38?

 We used the 19-37 or 20-38 epitopes (depending on the HPV serotype sequence annotations) since this epitope length was reported to be sufficient to elicit neutralizing antibodies (ref. 34). We have modified the sentence in the Introduction.

  • Lines 351-353, Figure 1F does not seem to support the conclusion that the OMV’s ‘had similar sizes’. The particles seem to range from 10 to 30 or 40 nm. Overall, the 8mer-OMVs appear to be smaller than the 4mer-OMVs. The authors should show size distributions, similar to Figure 5D, for the different OMVs.

Honestly, we thought that the visual inspection of the EM images, which include a 0-100 nm ruler, was sufficient to state that size and morphology do not substantially vary among OMVs preparation. This is supported the light scattering analysis reported in Figure 5D, showing that “empty” OMVs 4merB-OMVs are essentially identical in size distribution. We have now analyzed 8merAB-OMVs by light scattering and the size was about 45 +/- 5 nm.

  • Lines 368-371, Please, mention that number of animals per group was too small to draw statistically significant conclusion. The apparent differences between 4mers and 8mer at 10µg could also be due to biological variability in one or mice, or assay variability. Is the possible explanation for the slightly lover immunogenicity of the 8mer AB-OMVs correct? Based on the sizes of approx. 34kDa and 43kDa for the 4mer and 8mer fusion proteins respectively, the difference in total amount of protein would be approx. 25%. However, the 8mer contains twice as many cross-reactive L2 epitopes. So, one could also argue that the 8mer should be more immunogenic. Have the authors quantified the number of copies of the 4mers and 8mers that are present of the particles, e.g., by quantifying MUC1?

This reviewer is correct. At the end of the Section we have added the following sentence: “It has to be pointed out that the above conclusions have to be taken with a certain degree of caution since the number of animals per group is limited (4 mice) and because such conclusions do not take into consideration the possible difference in the expression level of the three constructs in the OMVs, expression which has not been rigorously quantified”.

  • Lines 410-420, Similar to the comment on lines 368-371, a higher number of fusion proteins on the 4merB-OMVs could also explain the neutralization data (Lines 410-420 and Figure 3), where 4merB, rather unexpectedly, outperforms 4merA-OMVs for the HPV types that are present in 4merA.

Again, this is correct. The following sentence has been added at the end of the paragraph: “Again, variability in fusion expression levels, which were not rigorously determined, could at least partially explain the difference among the neutralization titers of the three OMV vaccines”.  

  • Figure 4, represents data from Figures 2 and 3 in a different manner. Figure 4 should be moved to the Supplementary data and the description/conclusion (lines 433-438) can be shortened: ‘In general sera with the highest ELISA titers against a specific serotype also performed well in the neutralization assay and vice versa (Figure S…).

We agree. Figure 4 has been moved to Supplementary data (Figure S4) and we have shortened the sentence as proposed.

  • Lines 439-441, Delete sentences. In the absence of data, it is only an assumption that does not add value to the manuscript.

 We agree. The sentence has been deleted.

  • Lines 517-521, The prediction by Spagnoli et al. was based on sequence differences between HPV L2 epitopes, not on experimental data. They did not test different compositions in reference 41, they only compared HPV16 vs HPV-8x. Therefore, the conclusion that the OMVs might present the epitopes in a manner that leads to broader cross-reactivity (Lines 523-527) cannot be drawn. That would require a direct comparison between 4merB-OMVs and PfTrx-L2(18-33-35-39)-OVX313 nanoparticles.

 We have modified the sentence, which now reads: “Interestingly, on the basis of the sequence homology, Spagnoli et al. predicted that the L2 epitopes from the serotypes 16, 31, 51, 6, 18, 33, 35, and 59 were necessary and sufficient to design vaccines that could protect against more than 90% of all circulating oncogenic HPV and genital wart associated strains, including the serotypes which are the target of the current nonavalent Gardasil vaccine. Our neutralization data show that the expression of the 18-33-35-59 tetramer in OMVs (4merB) was sufficient to neutralize with high efficiency not only the homologous pseudovirus but also the 16, 31, 51, 6 pseudoviruses. Although the vaccine neutralization capacity has not been tested yet against other serotypes, on the basis of this result it is tempting to suggest that the L2 epitopes are presented on the surface of the OMVs in a configuration that allow the elicitation of antibodies with a cross-protective activity broader than what originally predicted”. Hope this is now acceptable.          

  • Lines 532-539, The inference that the authors try to make, is too speculative. The mechanisms of action of L1- and L2-based vaccines are probably different and mouse data cannot reliably predict (neutralizing) antibody titers in humans. Furthermore, the durability of the anti-L2 antibodies after vaccination with the OMV particles in humans is unknown. Proposed amendment of the text: ‘A relevant question is whether the neutralizing activity induced by our L2-expressing OMVs would be sufficient to protect HPV infection in humans and how the neutralization titers compare with the titers induced by the L1-based vaccines. Although head-to-head comparisons with commercially available vaccines in mice could be performed, the real evidence can only be obtained in humans. This said, the threshold of anti-L2 antibody titers for protection against HPV infection inn humans is not known.  

Thanks for the proposed suggestion. The paragraph has been modified as follows: “A relevant question is whether the neutralizing activity induced by our L2-expressing OMVs would be sufficient to protect HPV infection in humans and how the neutralization titers compare with the titers induced by the L1-based vaccines. Although head-to-head comparisons with commercially available vaccines in mice could be performed, the real evidence can only be obtained in humans. Romanowski et al.43 reported that in humans the HPV16 neutralizing antibody titers induced by the HPV-16/18 AS04-adjuvanted vaccine and measured 7 months after the third vaccination dose were in the range of 2 x 104. Should our OMV vaccine be in humans as immunogenic as in mice, the neutralization titers of the L2-based vaccine and the L1-based vaccine would not be too dissimilar. This said, the threshold of anti-L2 antibody titers for protection against HPV infection in humans is not known”. 

  • Lines 553-559, The authors suggestion to combine with existing L1-based vaccines contradicts their earlier statement that the L1-based vaccines are too expensive and dependent on the cold chain for transport and distribution. The authors should point out this contradiction or perhaps suggest to combine with E. colibased L1 vaccines.

 Again, this reviewer is correct. We have highlighted the disadvantage of the combination of the L1-based vaccines with our OMVs vaccine adding the following sentence: “Obviously, the limitation of this vaccine formulation would be that it does not overcome the problems of costs and stability of the existing vaccines, and therefore the use of such PAN-HPV vaccine risks to be restricted to high income countries”.  

  • Lines 585-587, The authors conclude that the combination with a tumor-specific epitope from MUC1 generates an interesting vaccine candidate to prevent HPV-associated tumors and they refer to the safety profile in a colon cancer study. However, is anything known about the safety of inducing anti-MUC1 antibodies in healthy people, the main target population of a preventative HPV vaccine?

 Yes, MUC1 vaccines have been given to healthy people considered to be at risk of adenoma recurrence. No safety issues have ever been reported (Finn, O, Nature Rev. Immunol. 2017).

Comments on the Quality of English Language

Minor comments

  • Line 37, delete ‘amazing’

Done.

  • Line 45-47, Suggestion to change to: ‘To develop cost-effective vaccines against infectious diseases and cancer, our laboratories have been working on a new vaccine platform based on engineered OMVs from non-pathogenic Escherichia coli (E. coli) derivatives over the last few years.

Thanks. Done

  • Table 1, move to supplementary materials.

Done

  • Lines 281-287, move to introduction.

With the permission of this reviewer, we prefer to keep the description of the experimental work in the Introduction Section as short as possible and therefore we would leave this sentence as it is.

  • Lines 293-301, partly repeating the description in Materials and Methods. Can be shortened substantially: ‘To test the feasibility of the approach, we first test expression of a single 293 L2 epitope, from HPV16, as a fusion protein with DomA and MUC1 (hereinafter DomA-L216) as described in Materials and Methods. As shown in Figure 1C, DomA-L216was expressed in the OMVs etc….’.

Done.

  • Line 302, delete ‘the’ in ‘…approximately the 24%’ of total OMV proteins…’.

Thanks. Done.

  • Line 303, ‘Moreover, the fusion protein was exposed….’.

Done.

  • Lines 322-326, refer to Materials and Methods instead of repeating the method. Note that the legend mentions 2% paraformaldehyde, while 4% is mentioned in the methods.

Done

  • Lines 400-401, A vaginal challenge model has been used to study HPV vaccine efficacy in mice. The authors should explain why they chose the in vitro neutralization assay.

As this reviewer knows very well. the vaginal challenge model is a good infection model but is not a true efficacy model. Therefore, as requested by the National and Local rules on animal experiments (3R rules), we opted for the use of an in vitro assay which similarly establishes the ability of antibodies to block pseudovirus entrance and at the same time avoids the sacrifice of a large number of animals.

  • Line 402, ‘According to In this assay,….’

Done

  • Line 406, Production of the pseudoviruses should be described in Materials and Methods (or a reference/source should be added).

We have described the production of the pseudoviruses in the Materials and Methods Section.

  • Lines 460-471, Delete sentences (already described in Materials and Methods).

Done.

Reviewer 3 Report

The existing HPV vaccines are only effective against the specific HPV types as they are based on L1 proteins. The authors describe a method of developing a pan-HPV effective vaccine based on the Minor Capsid Protein L2, using bacterial outer membrane vesicles as a platform for the creation of the L2 protein in abundance for immunisation. 

It is original and relevant to the field. It will revolutionise the method of producing antigens for not just HPV but many other other antigenic proteins from infectious diseases and, perhaps, cancers. It addresses the problem of producing a specific vaccine for each of the pathogenic variants of HPV.

The authors need to expand on this topic a bit more - is it likely to produce long lasting immunogenicity and protection? 

How does one proceed furhter to prove the immunogenecity, the induction of memory against HPV and the possible clinical use of the presented technology?

I am not sure the authors claim to have invented a new vaccine. They have only described a method of producing L2 antigen in large quantities in E Coli using the plasmids that will produce it through bacterial outer membrane vesicles - essentially, avoids using the same culture of E Coli endlesslessly, without killing the culture.

Kindly be brief with short sentences. Avoid colloquial expressions. 

Author Response

Reviewer 3

It is original and relevant to the field. It will revolutionise the method of producing antigens for not just HPV but many other other antigenic proteins from infectious diseases and, perhaps, cancers. It addresses the problem of producing a specific vaccine for each of the pathogenic variants of HPV. The authors need to expand on this topic a bit more - is it likely to produce long lasting immunogenicity and protection? 

Many thanks for the positive evaluation of our manuscript. In the Discussion Section We have further emphasized the advantage of our vaccine candidate, which avoids the preparation of different antigens (VLPs) to confer broad protection. As far as the duration of the immune response, please see below.

How does one proceed furhter to prove the immunogenecity, the induction of memory against HPV and the possible clinical use of the presented technology?

We believe we have sufficiently discussed the possible strategies for the clinical use of the OMV-based vaccine in the Discussion Section and we have added a sentence on how to follow the durability of the immune responses. Briefly, since the human population appears to be seronegative with respect to the L2 N-terminal epitope since neither natural infection nor L1 vaccination elicit antibodies against this particular L2, volunteers vaccinated with a L1 vaccine can receive our OMVs (4merB) vaccine and vaccine immunogenicity and the duration of anti-L2 responses are followed comparing anti-L1 and anti-L2 antibody titers with respect to control volunteers. Subsequently, OMV vaccine efficacy can be established following the incidence of infection/disease caused by non-L1 vaccine strains. Alternatively, our OMV vaccine could be combined with one of the existing L1 vaccines. Such combination would be extremely attractive since the anti-L1 and anti-L2 antibodies are expected to synergize. Moreover, the anti-L1 antibody titers should be further enhanced by the potent adjuvant contribution of the OMVs. Finally, considering the broad protective activity of the string of L2 epitopes, the vaccine combination is likely to become a universal PAN-HPV vaccine.

I am not sure the authors claim to have invented a new vaccine. They have only described a method of producing L2 antigen in large quantities in E Coli using the plasmids that will produce it through bacterial outer membrane vesicles - essentially, avoids using the same culture of E Coli endlesslessly, without killing the culture.

We have further strengthened this concept by adding the following sentence in the Introduction: “Such OMV-based vaccine is a novel effective vaccine which for the simplicity of its production process represents an effective and inexpensive solution to universal anti-HPV vaccination campaigns”.

Kindly be brief with short sentences. Avoid colloquial expressions.

Thanks. We have modified the manuscript trying to follow the recommendations.

Round 2

Reviewer 2 Report

I thank the authors for amending the manuscript